

# The epigenetic role of EZH2 in acute myeloid leukemia

Jinyong Fang[1], Jingcheng Zhang[1], Lujian Zhu[2], Xiaoru Xin[3] and Huixian Hu[1]

[1] Department of Hematology, Affiliated Jinhua Hospital, Zhejiang University School of Medicine, Jinhua, Zhejiang, China
[2] Department of Infectious Diseases, Affiliated Jinhua Hospital, Zhejiang University School of Medicine, Jinhua, Zhejiang, China
[3] College of Life Sciences, Zhejiang Normal University, Jinhua, Zhejiang, China

## ABSTRACT

Acute myeloid leukemia (AML), a malignant disease of the bone marrow, is characterized by the clonal expansion of myeloid progenitor cells and a block in differentiation. The high heterogeneity of AML significantly impedes the development of effective treatment strategies. Enhancer of zeste homolog 2 (EZH2), the catalytic subunit of the polycomb repressive complex 2 (PRC2), regulates the expression of downstream target genes through the trimethylation of lysine 27 on histone 3 (H3K27me3). Increasing evidence suggests that the dysregulation of EZH2 expression in various cancers is closely associated with tumorigenesis. In the review, we examine the role of EZH2 in AML, highlighting its crucial involvement in regulating stemness, proliferation, differentiation, immune response, drug resistance and recurrence. Furthermore, we summarize the application of EZH2 inhibitors in AML treatment and discuss their potential in combination with other therapeutic modalities. Therefore, targeting EZH2 may represent a novel and promising strategy for the treatment of AML.

## INTRODUCTION

Acute myeloid leukemia (AML), which represents the most prevalent type of leukemia in adults, is a malignant disease of the bone marrow characterized by the clonal expansion of myeloid progenitor cells as well as a block in their differentiation. Recent data indicate that globally, AML-related deaths exceed 80,000 each year, with projections suggesting that this number could double in the next 20 years (*DiNardo et al., 2023*). This disease predominantly affects the elderly, with a median diagnosis age of 68 years (*Song et al., 2018*). Therefore, as the population ages, AML is expected to place significant psychological and economic burdens on families and society. At the same time, despite advancements in prognostic markers and the development of various new drugs (both FDA-approved and in clinical trials), AML's mortality rate has unchanged over the past two decades. In fact, the overall five-year survival rate currently stands at 24.0%, and with a median overall survival of approximately 8.5 months, AML ranks fifth among the different cancer types (*Shallis et al., 2019*). This underscores the need to identify new and effective therapeutic targets to better diagnose and treat AML patients.

Corresponding authors
Xiaoru Xin, xinxiaoru@zjnu.edu.cn
Huixian Hu, huhuixian@zju.edu.cn

Typically, AML is driven by various genetic mutations, including FLT3, NPM1, DNMT3A, IDH1/IDH2, and KMT2A (MLL/MLL1). These mutations confer a growth advantage to specific clonal cells while blocking differentiation, which leads to disease progression (*Coombs, Tallman & Levine, 2016*; *Shih et al., 2012*). Concurrently, epigenetic alterations significantly contribute to the onset of AML. Dysregulation of DNA methylation and histone modifications often leads to transcriptional repression of key differentiation genes, causing AML cells to stagnate at early differentiation stages and preventing further maturation, thereby promoting leukemia development (*Wouters & Delwel, 2016*). Additionally, hypomethylating agents and histone deacetylase inhibitors have proven effective in treating AML and T-cell lymphomas, underscoring the essential role of epigenetic regulation in hematologic malignancies (*Zhao et al., 2023*). Therefore, a deeper understanding of the genetic and epigenetic mechanisms involved in AML is crucial for developing effective therapeutic strategies.

Histone modifications are a crucial component of epigenetics, playing a key role in the regulation of gene transcription. The main modifications include methylation, acetylation, phosphorylation, and ubiquitination. Notably, histone acetylation is typically associated with transcriptional activation, as it reduces the affinity between histones and DNA, allowing transcription factors and RNA polymerase easier access to the DNA. In contrast, methylation can either promote or inhibit transcription, depending on the specific sites and the number of methyl groups. For instance, dimethylation and trimethylation of H3K4, H3K36, and H3K79 are indicative of activation, while methylation of H3K9, H3K27, and H4K20 serves as marks of repressed and inactive genes (*Al Ojaimi et al., 2022*). Histone methylation is regulated by lysine methyltransferases (KMTs), arginine methyltransferases (PRMTs), and histone demethylases (HDMTs). In AML, KMT2A (also known as MLL) is a well-studied KMTs that primarily activates transcription of cell differentiation genes by catalyzing H3K4 trimethylation through its SET domain (*Xia et al., 2003*). However, MLL can fuse with multiple genes (such as AF9, AF4, AF10), leading to the aberrant activation of AML-related oncogenes (like HOXA9 and MEIS1), which enhances the proliferative capacity of AML cells (*Pastore & Levine, 2016*; *Tsai & So, 2017*). Conversely, rearranged MLL can lose its transcriptional activation function, resulting in halted cell differentiation and promoting leukemia development. Recently, EZH2, a prominent member of the KMTs family, has attracted considerable attention due to its role in myeloid malignancies. Therefore, this review will focus on the relevant studies of EZH2 in AML.

EZH2, a core catalytic component of the PRC2 complex, inhibits gene transcription by catalyzing the mono-, di-, or tri-methylation of H3K27. Current research indicates that EZH2 is involved in regulating various biological processes, including cell senescence, immune response, stem cell renewal and development as well as the development and malignancy of tumors. In hematopoiesis, EZH2 is critical for maintaining a balance between the self-renewal, differentiation and maturation of hematopoietic stem cells (HSCs) (*Kamminga et al., 2006*; *Mochizuki-Kashio et al., 2011*). Unsurprisingly, EZH2 is also involved in various hematological malignancies where it acts as an oncogene or a tumor suppressor. For instance, studies confirm that EZH2 possesses tumorigenic properties in lymphomas (*Van Kemenade et al., 2001*), chronic myelogenous leukemia (CML) (*Xie et*

*al., 2016*), multiple myeloma (*Xu et al., 2023*), natural killer/T-cell lymphoma (NKTL) (*Yan et al., 2013*) as well as AML. However, there are also reports of myelodysplastic syndromes (MDS) exhibiting mutations that inactivate EZH2 (*Nikoloski et al., 2010*), hence highlighting EZH2's potential to act as a tumor suppressor under certain cellular conditions. Although EZH2 is known to regulate gene expression through multiple mechanisms, its specific role in AML is yet to be established. Therefore, this review aims to comprehensively summarize existing research on EZH2 in the context of AML development, drug resistance and treatment. The goal is to enhance current understanding of the relationship between EZH2 and AML, while exploring EZH2's potential as a molecular target in view of improving the diagnosis and treatment of AML.

## THE INTENDED AUDIENCE AND NEED FOR THIS REVIEW

The development of acute myeloid leukemia (AML) involves a range of genetic and epigenetic alterations, including aberrant histone methylation. EZH2, a key catalytic enzyme in histone methylation, regulates the expression of relevant genes and consequently influences AML progression. Existing literature indicates that EZH2 inhibitors show promise in the treatment of AML. However, the detailed mechanisms of EZH2's role in AML remain unclear. Therefore, this review aims to summarize the research progress on the role of EZH2 in AML pathogenesis, drug resistance, and therapy. The objective is to enhance the understanding of the relationship between EZH2 and AML among scholars and clinical researchers in the field of hematological malignancies, while exploring the potential of EZH2 as a molecular target. Ultimately, this could lead to improved diagnosis and treatment of AML.

## SURVEY METHODOLOGY

To ensure an inclusive and unbiased analysis of the literature and to accomplish the review objectives, we searched the following literature databases: PubMed, Google Scholar and Web of Science. The search terms included: EZH2, PRC2, AML and epigenetics. It is worth noting that the keywords used and their variants and related words can be sorted, combined, and then searched.

This article is based on published literature. The aspects of the inclusion criteria are the retrieval keywords, information not covered by previous literature, and most importantly, the use of a clear and credible source.

### Structure and mechanism of EZH2

The *EZH2* gene, found on chromosome 7q36.1, encodes a 751 amino acids-long protein (*Han et al., 2007*). Structurally, EZH2 possesses ten domains, namely MCSS, SANT2, CXC and SET which represent the catalytic domains near the C-terminus as well as SANT1, SRM, SAL, BAM, EBD and SBD which are the regulatory domains near the N-terminus (*Wang et al., 2023a*) (Fig. 1). The SET and CXC domains are subsequently essential for histone methyltransferase activity (*Simon & Lange, 2008*). Specifically, the CXC domain mediates interactions with DNA and nucleosomes, while SET serves as the main site for methyltransferase activity and SAM (S-adenosyl-L-methionine) binding (*Jiao & Liu, 2015*).

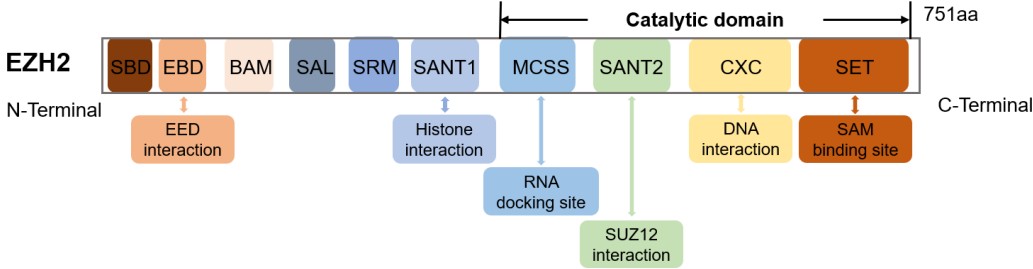

**Figure 1** The schematic diagram of the EZH2 domain composition along with their corresponding functions.

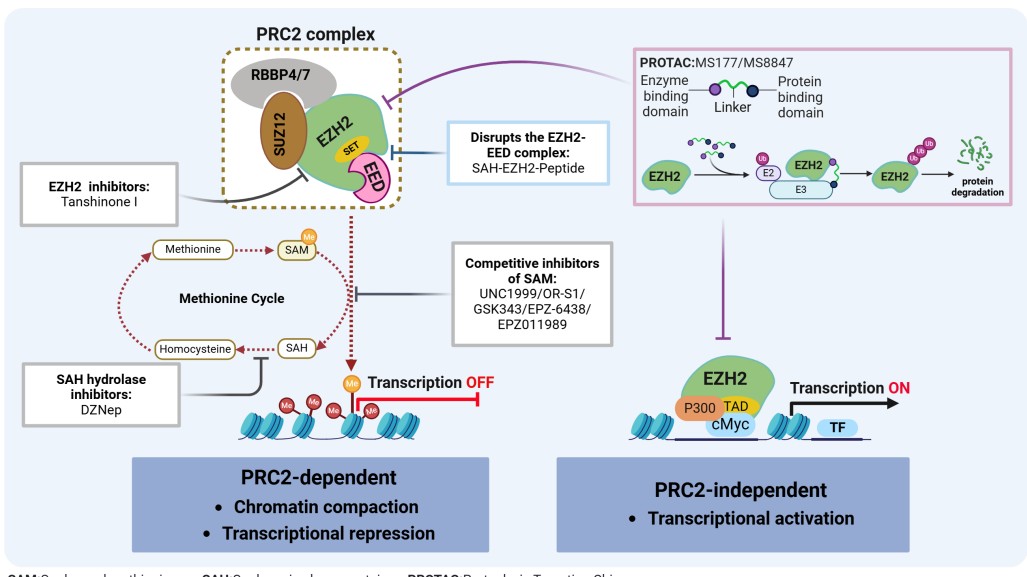

SAM: S-adenosyl methionine     SAH: S-adenosine homocysteine     PROTAC: Proteolysis Targeting Chimera

**Figure 2** **EZH2 regulates transcriptional activity in AML.** EZH2 regulates transcriptional activity through: (1) EZH2 induces H3K27 methylation *via* the PRC2 complex, leading to transcriptional repression; (2) EZH2 independently exerting a role in transcriptional activation. The relevant inhibitors of EZH2 and their mechanisms of action are described.

Typically, EZH2 cannot function independently but instead, can exhibit its methyltransferase activity when part of the PRC2 complex. This complex consists of EZH2, embryonic ectoderm development protein (EED), suppressor of zeste 12 (SUZ12), and the retinoblastoma-binding protein 4/7 (RBBP4/7) (Fig. 2). The canonical PRC2 targets histone 3, methylating H3K27 to produce H3K27me1, H3K27me2 and H3K27me3, with this process resulting in chromatin compaction and repressed transcription (*Bhattacharyya & Bond, 2023*). This process is essential for maintaining normal cellular functions and plays a pivotal role in embryonic development and stem cell maintenance. Conversely, the H3K27me3 demethylases, lysine demethylase 6A (KDM6A) and lysine demethylase 6B (KDM6B), dynamically regulate the methylation status of H3K27, ensuring the timely and accurate expression of genes (*Pastore & Levine, 2016*). The dynamic balance between H3K27

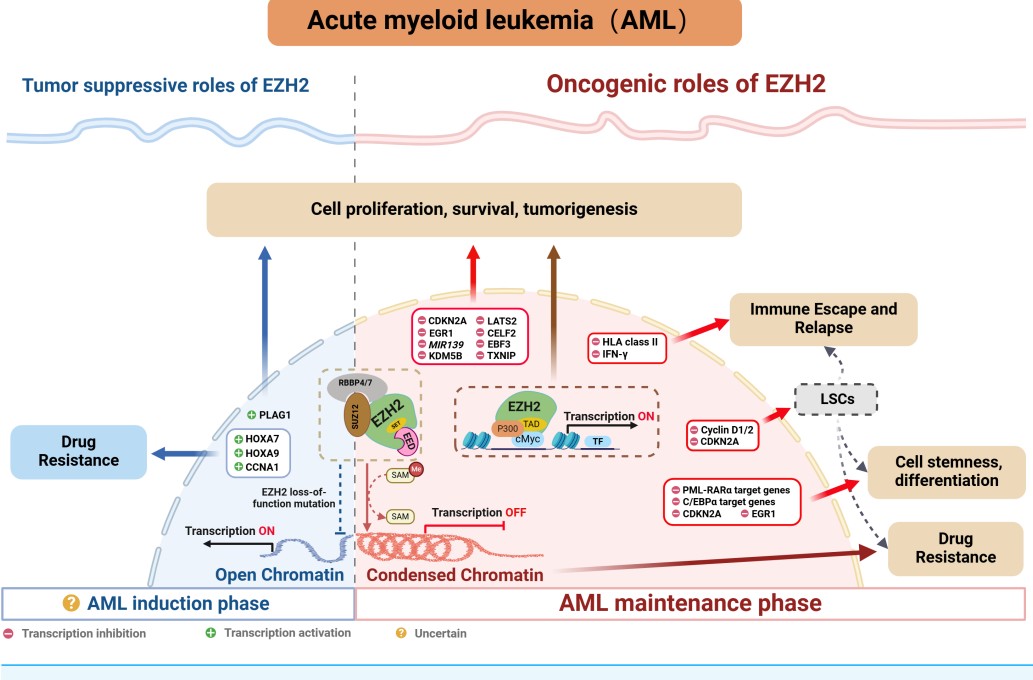

**Figure 3** EZH2's role in AML.

methylation and demethylation is crucial for normal cellular function. Disruption of this balance can lead to abnormal gene expression patterns, affecting stem cell differentiation and tissue development, ultimately contributing to tumorigenesis. Additionally, EZH2 can promote tumorigenesis, independently of the PRC2 complex, through non-canonical pathways such as the binding of cMyc and p300 to activate AML-related genes (*Wang et al., 2022*) (Fig. 2).

## EZH2's role in AML

Indeed, many studies confirm a close association between abnormal EZH2 expression and poor prognosis, resistance to chemotherapy as well as disease relapse in AML (Fig. 3). The following sections explore EZH2's relevance in AML, while providing a comprehensive summary of recent research on the subject.

## Oncogenic role of EZH2

EZH2 is typically regarded as being indispensable in AML, especially when mixed lineage leukemia (MLL) gene fusion is involved. Indeed, PRC2/EZH2 maintains the stem cell-like characteristics of MLL-AML cells by regulating their proliferation and differentiation (*Kim et al., 2013*; *Neff et al., 2012*; *Shi et al., 2013*; *Tanaka et al., 2012*). Furthermore, inhibiting EZH2 *in vivo* could not only reduce the frequency of leukemia-initiating cells (LICs) but also improve the survival of mice with MLL-AML (*Fujita et al., 2018*; *Ueda et al., 2014*). EZH2 has also been shown to be oncogenic in other AML subtypes. For instance, in WT1-mutated AML cells, using short hairpin RNA or pharmacological PRC2/EZH2 inhibitors was shown to promote myeloid differentiation (*Sinha et al., 2015*). Moreover, in studies

involving NUP98-NSD1$^+$ AML, the use of EZH2 enzymatic inhibitors demonstrated that tumorigenicity was dependent on PRC2/EZH2 (*Ren et al., 2022*). While clinical studies further confirmed the high expression of EZH2 in AML patients as well as its correlation with poor outcomes, including lower complete remission (CR) rates, higher rates of refractory/relapsed disease and shorter overall survival (OS) (*Yang et al., 2022*). In summary, the above results indicate that EZH2 is oncogenic in AML.

Interestingly, inactivating mutations of EZH2 have been reported in MDS and T-cell lymphoblastic leukemia (*Nikoloski et al., 2010*; *Ntziachristos et al., 2012*). Although such occurrences are relatively rare in AML, they have been documented (*Papaemmanuil et al., 2016*), seemingly contradicting EZH2's oncogenic role in AML. *Neff et al. (2012)* investigate this phenomenon and find that EZH2 loss weakens but does not completely eliminate leukemia growth. Genome-wide analysis of PRC2-mediated histone H3 trimethylation shows that H3K27me3 persists at specific loci despite EZH2 inactivation, indicating partial compensation by EZH1, as seen in embryonic stem (ES) cells (*Shen et al., 2008*). In contrast, EED inactivation results in a complete loss of PRC2 function, inhibiting leukemia cell growth. These findings highlight the critical role of PRC2 in AML and suggest that EZH1-mediated H3K27 methylation may compensate for EZH2 loss in rare cases.

EZH2 typically regulates tumor development through the classical pathway by forming the PRC2 complex which controls H3K27me3 deposition and represses transcription. EZH2 is currently known to regulate AML by repressing tumor suppressor genes such as *CDKN2A* (*Xu et al., 2015*), *EGR1* (*Tanaka et al., 2012*), *MIR139* (*Stavast et al., 2022*) and *TXNIP* (*Ueda et al., 2014*). In addition to that PRC2/EZH2 can inhibit downstream key targets such as KDM5B, with this process blocking the recruitment and/or assembly of oncogene suppression complexes (including HDAC-containing SIN3B and NuRD or other gene-repressive complexes) to maintain the oncogenicity of NUP98-NSD1$^+$ AML (*Ren et al., 2022*). On the other hand, EZH2 can act as a critical mediator in the tumorigenesis of AML. For instance, the long noncoding RNA HOXA-AS2 acts as an oncogene by binding with EZH2 and to suppress large tumor suppressor 2 (LATS2) in AML (*Feng et al., 2020*). Similarly, TNK2-AS1 can bind to EZH2 and silence CUGBP Elav-like family member 2 (CELF2), with these processes subsequently activating the PI3K/Akt pathway in AML cells to promote tumor development (*Guo et al., 2023*). Additionally, circadian clock protein BMAL1 enhances methylation of the early B-cell factor 3 (EBF3) promoter *via* EZH2 and suppresses its expression, inhibiting ferroptosis processes, thus promoting AML progression (*Wang et al., 2023b*). In addition to the above, EZH2 can also promote AML development, independently of the PRC2 complex, through non-canonical pathways. For example, the study by *Wang et al. (2022)* found that EZH2 uses a hidden transactivation domain (TAD) to interact with cMyc and p300, thereby activating AML-related transcripts and promoting oncogenesis. In revealing EZH2's oncogenic functions in AML as well as the possible underlying mechanisms, the above findings provide a strong basis for considering EZH2 as a potential target for AML therapy.

## Tumor suppressive role of EZH2

Chromatin regulators in cancer often have dual roles, acting as either tumor suppressors or promoters depending on cellular conditions. As far as EZH2 is concerned, its role in hematological malignancies is similarly complex and appears to be environment-dependent. In MDS, the presence of mutations that inactivate EZH2 suggested that, under certain cell conditions, EZH2 may act as a tumor suppressor. In AML, some studies also suggested that EZH2 may act as a tumor suppressor (*Nikoloski et al., 2010*). Low EZH2 expression was associated with poorer overall survival, event-free survival and relapse-free survival in AML patients (*Gollner et al., 2017*). In addition, among the relapsed AML samples, 45% of cases exhibited a loss of the EZH2 protein (*Gollner et al., 2017*; *Hamed et al., 2023*). Furthermore, in the French national ELAM02 study which involved 222 pediatric AML patients, the results suggested that mutations and deletions in the core components of PRC2 (EED, SUZ12 and EZH2) resulted in a subgroup of pediatric AML patients who presented a particularly aggressive disease (*Papaemmanuil et al., 2016*). Additionally, there is evidence of chemotherapy resistance in AML patients where EZH2 mutations led to a loss of the histone methyltransferase activity (*Kempf et al., 2021*). These reports highlight EZH2's potential to act as a tumor suppressor in AML. However, in *de novo* AML, loss-of-function mutations of EZH2 tend to be rare. Here, except for MDS-derived AML, an analysis of 714 patients found that the frequency of EZH2 mutations was only 1.8% in *de novo* AML (*Wang et al., 2013*), and this reflected uncertainties in the prevalence and prognostic value of EZH2 mutations in AML individuals.

Regarding the dual role of EZH2, *Basheer et al. (2019)* discovered that EZH2 could display completely opposite functions depending on the disease stage of AML. Specifically, during AML's early induction phase, EZH2 can act as a tumor suppressor, but in the established disease phase, it can facilitate disease progression. Indeed, following EZH2 excision, the expression profile during AML induction exhibited minimum overlap with that obtained from the maintenance phase of MLL-AF9 AML, with approximately 12% of genes (60 out of 496 genes) being common. Therefore, the distinct phenotypes could be attributed to the fact that different genes are derepressed during the two different AML phases (*i.e.,* induction and maintenance). Typically, PRC2/EZH2 complex is known to repress gene expression through the action of H3K27me3 marks on proximal promoters and distal enhancer cis-regulatory elements. During disease maintenance, the deletion of EZH2 results in a significant reduction of these H3K27me3 marks (*Tanaka et al., 2012*). Notably, during disease induction, the loss of EZH2 primarily exerts its tumor-suppressive effects by altering H3K27me3 at promoters rather than enhancers (*Basheer et al., 2019*). Additionally, the tumor-suppressive functions of EZH2 are to maintain repression of a small number of oncogenes, including pleiomorphic adenoma gene 1 (*Plag1*) and potentially *Lin28b*, which, upon derepression, will accelerate the development of AML. Interestingly, during MLL-AF9 maintenance, EZH2 loss does not upregulate Plag1 (*Basheer et al., 2019*). In the hematopoietic lineage, CRISPR/Cas9-mediated disruption of EZH2 significantly induced early AML, thereby providing further confirmation of the role of EZH2 in initiating AML (*Tulkens et al., 2023*). Therefore, further exploration of the detailed molecular mechanisms of EZH2 in AML will be crucial for guiding the precise use of EZH2 inhibitors in the future.

## Role of EZH2 in AML's stemness and differentiation

When differentiation is induced, CD34[+] hematopoietic progenitor cells (HPCs) as well as embryonic stem cells experience a significant phase where newly replicated chromatin lacks the repressive histone mark H3K27me3. This temporary 'open' chromatin state allows lineage-determining transcription factors (TFs) to bind to nascent DNA, thereby facilitating differentiation (*Petruk et al., 2017a*; *Petruk et al., 2017b*). However, in AML blasts, the rapid accumulation of H3K27me3 on nascent DNA impedes cell differentiation (*Porazzi et al., 2022*). The levels of H3K27me3 largely depend on EZH2 activity, with numerous studies showing that inhibiting or deleting EZH2 can induce myeloid differentiation in AML cells. For example, in acute promyelocytic leukemia (APL), disrupting the PRC2/EZH2 complex can not only reverse histone modifications but also induce DNA demethylation of PML-RARα target genes, which, in turn, reactivates differentiation-related genes and promotes cell differentiation (*Villa et al., 2007*). Similarly, in non-APL AML, specific inhibition of EZH2 can also enhance differentiation (*Sbirkov et al., 2023*; *Sinha et al., 2015*; *Sung et al., 2024*). Beyond regulating cell differentiation, EZH2 is also crucial for maintaining the stemness of AML cells (*Fujita et al., 2018*; *Kikushige et al., 2023*; *Ueda et al., 2014*).

Mechanistically, EZH2 promotes the trimethylation of H3K27 at cell cycle-related genes, such as *Cyclin D1/2* (*Fujita et al., 2018*) and *CDKN2A* and at differentiation-related genes such as *EGR1* (*Tanaka et al., 2012*). The trimethylation process inhibits the expression of these genes, thereby blocking the differentiation program of leukemic stem cells while maintaining their stemness (*Tanaka et al., 2012*). Additionally, *Thiel et al. (2013)* found that EZH2 acts alongside menin to epigenetically suppress the expression of pro-differentiation C/EBPα target genes and block the mature differentiation of MLL-AF9[+] leukemia cells.

## Role of EZH2 in AML's drug resistance

Studies have established a close link between PRC2/EZH2 and chemotherapy resistance in AML. This correlation can be attributed to H3K27me3-marked nucleosome arrays which form the most compact chromatin structures in the genome (*Margueron & Reinberg, 2011*; *Yuan et al., 2012*), thereby reducing the effectiveness of DNA-damaging agents (*Pang et al., 2015*; *Pang et al., 2013*). In this context, inhibiting EZH2 in cell lines, primary cells and xenograft mouse models to de-condense the H3K27me3-marked chromatin of AML cells was shown to improve accessibility to chromatin and suppress leukemia through chemotherapy-induced DNA damage as well as apoptosis (*Porazzi et al., 2022*). Based on the above, researchers began exploring the combination of EZH2 inhibitors and DNA-damaging cytotoxic agents in AML treatment, and their findings confirmed that EZH2 inhibition could enhance DNA damage by cytotoxic drugs (such as doxorubicin and cytarabine) (*Fujita et al., 2018*; *Porazzi et al., 2022*). On the other hand, leukemia stem cells (LSCs), with their characteristics of self-renewal and quiescence, are key contributors to chemotherapy resistance and relapse in AML (*Gentles et al., 2010*; *Ishikawa et al., 2007*; *Kreso & Dick, 2014*). According to reports, quiescent AML cells exhibit the highest EZH2 expression, and combining an EZH1/2 dual inhibitor with cytarabine could significantly reduce the number of leukemic granulocyte macrophage progenitors (L-GMPs), with this cell population nearly disappearing (*Fujita et al., 2018*). These studies indicate that

inhibiting EZH2 could represent a promising strategy for overcoming chemotherapy resistance in AML.

Similar to the dual role of EZH2 in cancer pathogenesis, existing research indicates that the loss of EZH2 can also lead to drug resistance in AML. *Rathert et al. (2015)* noted that inhibiting the PRC2/EZH2 complex could induce resistance to BET inhibitors by remodeling regulatory pathways, thereby restoring the transcription of key targets such as MYC. Additionally, the activation of *HOXA7* and *HOXA9* in response to the absence of endogenous EZH2 is crucial for promoting resistance to tyrosine kinase inhibitors and other cytotoxic drugs in AML cells (*Gollner et al., 2017*). Furthermore, researchers found that the loss of EZH2 reduced H3K27me3 modification on the CCNA1 promoter, thereby enhancing CCNA1 expression as well as AML cell resistance to drugs such as PKC412, AC220 and AraC (*Yang et al., 2021*). These findings suggest that restoring EZH2 function could be a viable strategy for overcoming treatment resistance in AML patients.

## Role of EZH2 in AML's immune escape and relapse

Over the past few years, significant advancements have enhanced the feasibility and safety of allogeneic hematopoietic stem cell transplantation (allo-HSCT) for curing AML. However, post-transplantation relapses, which affect up to 50% of patients, remain a major challenge. In this context, it is now widely recognized that immune evasion is the primary driver of such relapses (*Zeiser & Vago, 2019*), with subsequent research showing that, in leukemia cells, nongenomic loss of HLA class II expression could account for up to 40% of the relapse cases (*Christopher et al., 2018*; *Toffalori et al., 2019*). *Gambacorta et al. (2022)* recently reported that the loss of HLA class II expression was associated with PRC2/EZH2-dependent reductions in chromatin accessibility. Reducing the repression of HLA class II genes by PRC2/EZH2 could enhance the immunogenicity of relapsed leukemia, thereby improving T cell-mediated recognition and killing. This highlights PRC2/EZH2 as a key epigenetic factor in this immune escape mechanism. In addition, PRC2/EZH2 is known to regulate the expression of HLA and other immune-related molecules in solid tumors and lymphomas, thus underscoring its role as a potent regulator of antigen presentation (*Burr et al., 2019*; *Dersh et al., 2021*; *Ennishi et al., 2019*). Additionally, PRC2/EZH2 inhibitors induce the release of IFN$\gamma$ by CD4$^+$ T cells (*Gambacorta et al., 2022*), with the subsequent exposure of leukemic cells to IFN$\gamma$ restoring HLA class II expression to enhance AML's immunogenicity (*Canadas et al., 2018*; *Morel et al., 2021*). These findings suggest that targeting PRC2/EZH2 can be a promising approach to overcome re-establish an effective graft-versus-leukemia effect.

Relapse after achieving remission of AML is often caused by a few therapy-resistant cells within minimal residual disease (*Srinivasan Rajsri, Roy & Chakraborty, 2023*). These resistant cells, known as LSC, possess long-term self-renewal capacity which can promote clonal outgrowth and, as such, they are generally considered to be the initiating cells for tumor relapse (*Gentles et al., 2010*; *Ishikawa et al., 2007*; *Kreso & Dick, 2014*; *Stelmach & Trumpp, 2023*). Notably, EZH1/2 expression is highest in LSCs during their quiescent state (*Fujita et al., 2018*). Such elevated levels of EZH1/2 inhibit the expression of cell cycle protein D, resulting in the hypo-phosphorylation of the tumor suppressor gene Rb. This

**Table 1  EZH2 inhibitors and their applications in AML.**

| Inhibitors | Mechanism | Experimental model | Downstream targets | Biological effect | Reference |
|---|---|---|---|---|---|
| DZNep | S-adenosylhomocysteine hydrolase inhibitors | *In vitro* AML cell lines & LSCs | *CDKN2A, CDKN1A,* p27, *FBXO32,* Cyclin E, *HOXA9, TXNIP* | Promotes apoptosis & reduces LSCs | *Fiskus et al. (2009), Ueda et al. (2014)* and *Zhou et al. (2011)* |
| UNC1999 | Competitive inhibitors of S-adenosylmethionine | *In vitro* MLL-AML & murine leukemia model | *CDKN2A* | Inhibits growth & prolongs murine survival | *Xu et al. (2015)* |
| OR-Sl | Competitive inhibitors of S-adenosylmethionine | *In vitro* AML cell lines & murine leukemia model | Cyclin D1/2, Phospho-Rb | Inhibits growth & induces differentiation and apoptosis & prolongs murine survival | *Fujita et al. (2018)* and *Honma et al. (2017)* |
| GSK343 | Competitive inhibitors of S-adenosylmethionine | Kasumi-1& Murine leukemia model & AML patient samples | *CDKN1A, NFKBIZ* | Promotes apoptosis and arrests G0/G1 cell cycle | *Basheer et al. (2019)* |
| EPZ-6438/ EPZ011989 | Competitive inhibitors of S-adenosylmethionine | *In vitro* AML cell lines & AML1-ETO9a secondary leukemic mice | *CD38* | Induces differentiation & decreases colony-forming capabilities & prolongs murine survival | *Basheer et al. (2019)* and *Fobare et al. (2024)* |
| Tanshinone I | Direct binding to EZH2 | *In vitro* AML cell lines & zebrafish transgenics and xenograft models | *MMP9, ABCG2* | Induces differentiation and apoptosis | *Huang et al. (2021)* |
| HKMTI-1-005 | Competitive inhibitors of amino acids N-terminal of the substrate lysine residue | *In vitro* AML cell lines | *CD36, ITGAM* | Induces differentiation | *Sbirkov et al. (2023)* |
| MS177/MS8847 | EZH2 PROTAC degrader | *In vitro* AML cell lines & PDX animal model of MLL-AML | EZH2, cMyc | Decreases colony-forming capabilities & arrests cell cycle & induces apoptosis | *Velez et al. (2024)* and *Wang et al. (2022)* |
| SAH-EZH2 Peptides | Disrupts the EZH2-EED complex and reduces EZH2 protein levels | *In vitro* MLL-AF9 cells & mouse myeloid leukemia cell lines | EZH2, *ADAM8, p19ARF, ACE, CD133* | Inhibits growth & arrests cell cycle & induces differentiation | *Kim et al. (2013)* |

inhibition subsequently prevents cells from entering the S phase, thereby maintaining LSCs in a quiescent state (*Fujita et al., 2018*). These results suggest that the dysregulated expression of EZH2 plays a crucial role in AML relapse by modulating the behavior of LSCs.

## Targeting EZH2 drugs and strategy in AML

Given the role of EZH2 in the biology of AML diseases as well as its enzymatic activities, targeting EZH2 can be a promising approach for treating AML. In fact, in preclinical models, various compounds have already demonstrated potential antileukemic activities by disrupting the PRC2/EZH2 complex and/or downregulating its enzymatic functions (Table 1 & Fig. 2). Furthermore, despite the dual role of EZH2 in initiating and maintaining the disease, evidence consistently suggests that EZH2 is functionally required for full-blown AML, thus making it an attractive therapeutic target.

However, the efficacy of epigenetic-targeted therapies has been disappointing in some studies, especially when applied as monotherapies. This has fueled efforts for their integration into combination therapies (*Kim et al., 2013*) since combining EZH2 inhibitors with other therapeutics, such as the pan-histone deacetylase inhibitor panobinostat (*Fiskus et al., 2009*), 5-Azacytidine (*Momparler et al., 2012*), the BCL-2 inhibitor Venetoclax (*Yang et al., 2022*), FLT3 inhibitors (*Sung et al., 2024*), doxorubicin (*Porazzi et al., 2022*), Ara-C (*Fujita et al., 2018*) and ATRA (*Sbirkov et al., 2023*), can improve the efficacy of AML treatment and overcome the limitations of monotherapy. In particular, combining DZNeP with venetoclax was shown to significantly eliminate CD117$^+$ (c-KIT) AML blasts, hence indicating substantial effects on tumor stem cells (*Momparler et al., 2012*). Similarly, combining vitamin C with DZNeP enhanced apoptosis and differentiation in AML cell lines, while slowing leukemia progression *in vivo* (*Long et al., 2022*). At the molecular level, vitamin C exerts a synergistic anti-AML effect by downregulating the serine synthesis enzyme PHGDH as well as the anti-apoptotic gene *BCL-2* (*Long et al., 2022*). However, clinical trials are still required to further assess the effects of multi-drug combinations, especially with patients' tolerance and side effects being significant challenges.

## CONCLUSION AND PERSPECTIVE

In AML, EZH2-related epigenetic modifications were shown to play crucial roles in regulating cell stemness, differentiation and proliferation as well as contributing to AML drug resistance and relapse. The classical PRC2-dependent H3K27me3 modification represents the primary mechanism through which EZH2 silences the transcription of relevant genes in AML. However, under certain conditions, non-classical mechanisms, such as transcriptional co-activation, may also contribute significantly to disease development. Additionally, recent studies have shown that EZH2's functions can vary dramatically depending on the disease stage in AML and the cellular context. For instance, EZH2 acts as a tumor suppressor during AML's induction phase but promotes the disease in established AML (*Basheer et al., 2019*). However, this theory does not fully explain the intricate mechanisms of EZH2 in AML, indicating that further research is required.

Although EZH2 exerts dual functions in disease initiation and maintenance, evidence suggests that it is crucial for the progression of full-blown AML, thus making it an attractive therapeutic target in the treatment of AML. In summarizing current research on various EZH2 inhibitors used in AML treatment (Table 1), this review discusses the potential of combining these inhibitors with immunotherapy, conventional chemotherapy, targeted therapy and other treatment methods for AML. Interestingly, both mono- and combination therapies involving EZH2 inhibitors have shown potential in enhancing the efficacy of AML treatment. However, the efficacy of EZH2 inhibitors is still at the preclinical stage, and their effectiveness and safety profile require further evaluation in clinical trials.

### Funding

This review was supported by the Zhejiang Provincial Natural Science Foundation of China (LTGY24H080009), the Jinhua Municipal Science and Technology Bureau Key Project (2021-3-156), and the Jinhua Municipal Central Hospital Science and Technology Project (JY2022-5-01). The funders had no role in study design, data collection and analysis, decision to publish, or preparation of the manuscript.

### Grant Disclosures

The following grant information was disclosed by the authors:
Zhejiang Provincial Natural Science Foundation of China: LTGY24H080009.
Jinhua Municipal Science and Technology Bureau Key Project: 2021-3-156.
Jinhua Municipal Central Hospital Science and Technology Project: JY2022-5-01.

### Competing Interests

The authors declare there are no competing interests.

### Author Contributions

- Jinyong Fang conceived and designed the experiments, performed the experiments, analyzed the data, prepared figures and/or tables, authored or reviewed drafts of the article, and approved the final draft.
- Jingcheng Zhang performed the experiments, analyzed the data, prepared figures and/or tables, and approved the final draft.
- Lujian Zhu performed the experiments, analyzed the data, prepared figures and/or tables, and approved the final draft.
- Xiaoru Xin conceived and designed the experiments, performed the experiments, authored or reviewed drafts of the article, and approved the final draft.
- Huixian Hu conceived and designed the experiments, performed the experiments, authored or reviewed drafts of the article, and approved the final draft.

### Data Availability

This is a literature review.

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
