# Peer review of "The epigenetic role of EZH2 in acute myeloid leukemia"

_PeerJ, doi:10.7717/peerj.18656_

## Round 0.1 · original submission · Minor Revisions

Thank you for submitting your manuscript to PeerJ. I am sorry for the delay in the review process due to the unavailability of the reviewers. Three reviewers have reviewed it, and their comments can be seen below. Although the work is interesting, all reviewers suggested minor revisions before it can be published. I look forward to receiving the revised manuscript soon.

Reviewer 1 ·

Basic reporting

The current review is well structured and summarizes the key components of EZH2 in AML. The authors ensured to include as much of the current literature as possible in the review which ensures a comprehensive overview of the current data is available.

The justification for the review is established early on in the article, and the review is written in clear and professional English, however the abstract could benefit from further grammatical editing. Few grammatical errors exist in text which is addressed in the PDF attached comments of the review.

The authors have provided adequate references to support the findings of the review.

The review seems to be relevant, specifically with its focus on AML.

Experimental design

The study design is in line with the journal guidelines.
The authors were unbiased in their methodology approach and discusses advantages and limitations of EZH2 and its epigenetic role in AML. All sources are well cited, and the reviews follows a chronological design with subheadings.

Validity of the findings

The authors address the topic adequately and their findings provides much required inside in a summarized approach to the potential therapeutic role of EZH2 inhibitors in AML

Additional comments

Additional comments can be found in the attached PDF of the review.

Annotated reviews are not available for download in order to protect the identity of reviewers who chose to remain anonymous.
Cite this review as

Reviewer 2 ·

Basic reporting

No

Experimental design

No

Validity of the findings

No

Additional comments

Spelling and grammatical errors present throughout the manuscript that need correction. Some examples are given below:
1. In the first line of the abstract: Acute myeloid leukemia (AML), a malignant disease of the bone marrow, ‘is’ characterized by the clonal expansion of myeloid progenitor cells and a block in differentiation.
2. Line 28 remove ‘role’ from the sentence.
3. Line ‘31’ use ‘and’ instead of ‘but’.
4. Line 52 remove ‘as’

Other comments:
1. In the context of EZH2 mutations, the role of its paralog EZH1 should be discussed.
2. A separate section describing the EZH2 alterations, and the mechanism of this dysregulation should be included.
3. Since EZH2 is a H3K27 methyltransferase, a section highlighting the importance of this modification, along with KDM6, its demethylase, will improve understanding of their dynamism and overall impact.

Cite this review as

Reviewer 3 ·

Basic reporting

no comment, as this is a review article

Experimental design

no comment, as this is a review article

Validity of the findings

no comment, as this is a review article

Additional comments

In this paper Fang et al. review “The epigenetic role of EZH2 in acute myeloid leukemia.” This is a timely topic, with their being great interest in epigenetic regulators and “epi-drugs” in AML. After a brief introduction on AML, the authors focus first on the biology and biochemistry and EZH2, and then cover the opposing roles of EZH2 as a context-dependent oncogene as well as tumor suppressor gene in AML. The authors explore some of the biology underlying these roles and then move into a discussion of the role of EZH2 in drug resistance and immune escape. They conclude with commentary on drugs targeting EZH2.

This is altogether a helpful review, though several areas need improvement. Specifically –

- The introduction and background on AML is very thin. To properly understand the context of EZH2 in AML, the authors need to cover more of the basics of AML, i.e. how it is driven by clonal expansion and characterized by differentiation arrest, and how the epigenetic programs are known to be awry.

- There is no background on epigenetics in general. Perhaps even a very cursory review of the role of histone modifications on transcriptional processes would be helpful.

- Two seemingly contradictory points are made: (1) on lines 114-115 it is stated that high expression of EZH2 in AML correlates with poor outcomes, while (2) on line 143 it is stated that low EXH2 expression was associated with poorer survival. Could the authors provide any insight as to why these studies came to such opposite conclusions?

- Regarding the opposing roles of EZH2 as an oncogene and tumor suppressor gene (TSG) in AML, could the authors go just a bit more in-depth into the models of how and why different sets of genes are targeted by EZH2 in each case?

- One of the biggest issues is that, throughout the manuscript, processes and genes are mentioned without explaining what they are, which will make it difficult for the reader. Just a couple examples:

o In just lines 126-132, processes/genes that are not defined include “LATS2,” “CELF2,” “ferroptosis,” and “hidden TAD.”
o On line 165 a “feto-oncogenic program that involves genes such as Plag1” it will not be clear to most readers what this means
- In the section starting on line 171 the authors mention how CD34+ cells experience a temporary “open” chromatin phase that allows TFs to bind DNA, but that AML blasts do not have this stage and thus preclude the binding of these factors. Could the authors provide any information about why this would be, i.e. why do the regular blood cells have the open chromatin phase while AML blasts do not?

- On line 184 it is said that EZH2 methylates Cyclin D1/2 and CDKN2A as well as differentiation genes such as EGR1 to block differentiation and maintain stemness of leukemia stem cells. Why would repressing cell cycle genes such as Cyclin D1/2 help block differentiation?

Cite this review as

---

## Round 0.2 · accepted · Accept

The authors have addressed all the reviewers' comments, and all reviewers recommended that the manuscript be published.

Reviewer 1 ·

Basic reporting

N/A

Experimental design

N/A

Validity of the findings

N/A

Additional comments

The authors have addressed the previous comments. I therefore propose to accept the manuscript for publication.

Cite this review as

Reviewer 2 ·

Basic reporting

no

Experimental design

no

Validity of the findings

no

Additional comments

All concerns have been addressed.

Cite this review as

Reviewer 3 ·

Basic reporting

no comment (this is a review article)

Experimental design

no comment (this is a review article)

Validity of the findings

no comment (this is a review article)

Additional comments

The revised version of the manuscript is greatly improved. The concerns I raised have been addressed.

Cite this review as